# Two-Dimensional Weisfeiler-Lehman Graph Neural Networks for Link Prediction

## Abstract

Link prediction is one important application of graph neural networks (GNNs). Most existing GNNs for link prediction are based on one-dimensional Weisfeiler-Lehman (1-WL) test. As pointed out by previous works, 1-WL-GNNs by nature learn node-level representations thereby have poor expressive power on links. Some node labeling methods relieve this weakness but introduce low efficiency. In this paper, we study a completely different approach which directly obtain node pair (link) representations based on *two-dimensional Weisfeiler-Lehman (2-WL) tests*. 2-WL tests directly use links (2-tuples) as message passing units instead of nodes, and thus can directly obtain link representations. We theoretically analyze the expressive power of 2-WL tests to discriminate non-isomorphic links, and prove their superior link discriminating power than 1-WL. Based on different 2-WL variants, we propose a series of novel 2-WL-GNN models for link prediction. Experiments on a wide range of real-world datasets demonstrate their competitive performance to state-of-the-art baselines.

## 1 Introduction

Link prediction is a key problem of graph-structured data (Al Hasan et al., 2006; Liben-Nowell & Kleinberg, 2007; Menon & Elkan, 2011; Trouillon et al., 2016). It refers to utilizing node characteristics and graph topology to measure how likely a link exists between a pair of nodes. Due to the importance of predicting pairwise relations, it has wide applications in various domains, such as recommendation in social networks (Adamic & Adar, 2003), knowledge graph completion (Nickel et al., 2015), and metabolic network reconstruction (Oyetunde et al., 2017).

One class of traditional link prediction methods are heuristic methods, which use manually designed graph structural features of a target node pair such as number of common neighbors (CN) (Liben-Nowell & Kleinberg, 2007), preferential attachment (PA) (Barabási & Albert, 1999), and resource allocation (RA) (Zhou et al., 2009) to estimate the likelihood of link existence. Another class of methods, embedding methods, including Matrix Factorization (MF) (Menon & Elkan, 2011) and node2vec (Grover & Leskovec, 2016), learn node embeddings from the graph structure in a transductive manner, which cannot generalize to unseen nodes or new graphs. Recently, with the popularity of GNNs, their application to link prediction brings a number of cutting-edge models (Kipf & Welling, 2016; Zhang & Chen, 2018; Zhang et al., 2021; Zhu et al., 2021).

Most existing GNN models for link prediction are based on one-dimensional Weisfeiler-Lehman (1-WL) test (Weisfeiler & Leman, 1968; Shervashidze et al., 2011). 1-WL test is a popular heuristic for detecting non-isomorphic graphs. In each update, it obtains all nodes' new colors by hashing their own colors and multisets of their neighbors' colors. Vanilla GNNs simulate 1-WL test by iteratively aggregating neighboring node features to the center node to update node representations, which we call 1-WL-GNNs. With the node representations, 1-WL-GNNs compute link prediction scores by aggregating pairwise node representations. Graph Auto-encoder (GAE, and its variant VGAE) (Kipf & Welling, 2016) is such a model. However, 1-WL-GNNs can only discriminate links on the "node" level. This is illustrated by Figure 1 left: $v_2$ and $v_3$ are symmetric nodes in the graph thus having the same representation by 1-WL-GNN, but links $(v_1, v_2)$ and $(v_1, v_3)$ are not symmetric. However, 1-WL-GNNs are unable to discriminate links $(v_1, v_2)$ and $(v_1, v_3)$, though $(v_1, v_2)$ has a shorter path between them than $(v_1, v_3)$. Although positional node embeddings or random features can alleviate

this problem, they fail to guarantee symmetrical links (such as $(v_1, v_2)$ and $(v_4, v_3)$) to have the same representation.

In order to surpass 1-WL, plenty of link prediction models apply node labeling inherently, including SEAL (Zhang & Chen, 2018), Distance Encoding (Li et al., 2020), ID-GNN (You et al., 2021), and some models for matrix completion (Zhang & Chen, 2020) and knowledge graph completion (Teru et al., 2020). It raises expressive power from "node" to "link" level by breaking the symmetry between the target node pair and other nodes during the message passing. Figure 1 middle and right illustrated this effect. However, labeling also introduces a challenge. It requires repeatedly applying GNN to a labeled subgraph for **every** link to predict thereby being inefficient. Therefore, we aim to develop novel GNN models with both full-batch link prediction ability and higher expressive power than 1-WL.

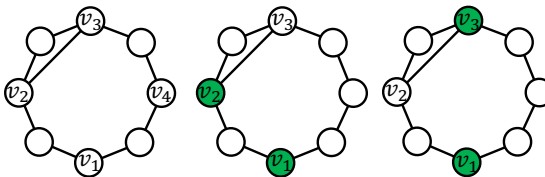

Figure 1: 1-WL-GNNs cannot distinguish links $(v_1, v_2)$ and $(v_1, v_3)$ in the left graph. With labeling trick, 1-WL-GNNs can distinguish them in their respective labeled graphs (middle and right).

We propose a completely different paradigm for link prediction. We construct GNNs based on two-dimensional Weisfeiler-Lehman (2-WL) tests, which we call 2-WL-GNNs. In 2-WL-GNNs, node pairs are used as the elemental message passing units so that link representations are directly obtained. Figure 2 gives an illustration for a particular 2-WL algorithm. We first theoretically study the link discriminating power of different 2-WL test variants, including the plain 2-WL, 2-FWL (Folklore WL), and their newly proposed local version. We show that 2-WL, 2-FWL and local 2-FWL are strictly more expressive than 1-WL for link prediction, while local 2-WL has equivalent power to 1-WL. Based on these 2-WL tests, we construct a series of 2-WL-GNN models. Despite all using node pairs to propagate messages, these models have different aggregation schemes, link discriminating power, time/space complexity, as well as drastically different implementations, which we discuss in Section 4. Extensive experiments on multiple benchmark datasets verify 2-WL-GNNs' power for link prediction. 2-WL-GNNs achieve highly competitive link prediction performance to state-of-the-art models including SEAL (Zhang & Chen, 2018) and NBFNet (Zhu et al., 2021), while using significantly less time.

In this paper, we aim to develop novel GNN models with both full-batch link prediction ability and higher expressive power than 1-WL.

## 2 LINK-LEVEL TWO-DIMENSIONAL WEISFEILER-LEHMAN TESTS

In this section we introduce various 2-WL tests which directly use links as message passing unit, and define their link-level expressive power. We denote a set by $\{\cdot\}$, an ordered set (tuple) by $(\cdot)$ and a multiset by $\{\!\!\{\cdot\}\!\!\}$ to have repeated elements. We use $[n]$ to denote the set $\{1, 2, ..., n\}$.

### 2.1 $k$-DIMENSIONAL WEISFEILER-LEHMAN TESTS

$k$-dimensional WL test ($k$-WL) uses $k$-tuples of nodes as update unit. In each iteration, every $k$-tuple updates its color from its newly-defined neighboring $k$-tuples. There are two variants of $k$-WL algorithms: the plain $k$-dimensional WL ($k$-WL) and the $k$-dimensional Folklore WL ($k$-FWL) (Cai et al., 1992; Grohe, 2017; Maron et al., 2019). Both $k$-WL and $k$-FWL update colors for $k$-tuples $\boldsymbol{s} := (s_1, s_2, ..., s_k)$ with $s_1, ..., s_k$ being nodes.

$k$-WL defines neighborhood of $k$-tuple $\boldsymbol{s}$ as $N(\boldsymbol{s}) = \big(N_1(\boldsymbol{s}), N_2(\boldsymbol{s}), ..., N_k(\boldsymbol{s})\big)$, where

$$N_j(\boldsymbol{s}) = \{\!\!\{(s_1, ..., s_{j-1}, s', s_{j+1}, ..., s_k)|s' \in [n]\}\!\!\}. \tag{1}$$

$k$-FWL has a different definition of neighborhood. $k$-FWL defines the $j$th neighborhood of $\boldsymbol{s}$ as

$$N_j^F(\boldsymbol{s}) = \big((j, s_2, ..., s_k), (s_1, j, ..., s_k), ..., (s_1, ..., s_{k-1}, j)\big). \tag{2}$$

And the full neighborhood of $\boldsymbol{s}$ is given by $N^F(\boldsymbol{s}) = \{\!\!\{N_j^F(\boldsymbol{s})|j \in [n]\}\!\!\}$. Essentially, $k$-WL and $k$-FWL have the same $nk$ neighbor tuples but differ in how these $nk$ tuples are **ordered and grouped**.

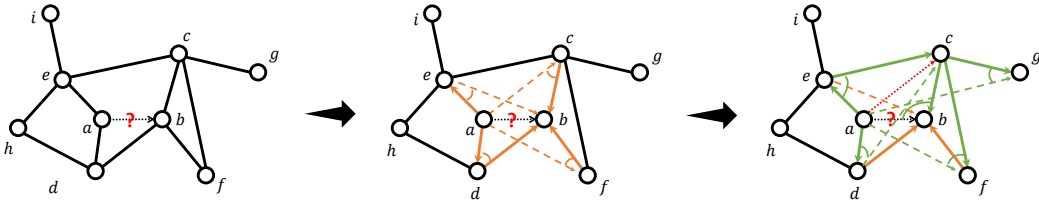

Figure 2: This figure illustrates how local 2-FWL works for link prediction. It takes links as message passing units. Given link $(a, b)$ to predict, it aggregates four orange link pairs in the first iteration (middle), where common neighbor $d$ is learned. Then in the second iteration (right), it aggregates five green link pairs through link $(a, c)$, where a 3-path $(a, e, c, b)$ between $a, b$ is captured.

They result in different expressive power between $k$-WL and $k$-FWL. In previous work $k$-WL and $k$-FWL's discriminating power for **graphs** has been studied that $k$-FWL has equal graph discriminating power to $(k + 1)$-WL which is strictly stronger than $k$-WL for $k \geq 2$ (Cai et al., 1992; Grohe, 2017).

## 2.2 VARIANTS OF LINK-LEVEL WEISFEILER-LEHMAN TESTS

Link-level tasks require representation of one or several links within the whole graph. It is natural to investigate the $k = 2$ case as link representation can be directly encoded by its color in test. We use $c^{(t)}(e)$ to denote the color of link $e := (p, q) \in [n] \times [n]$ at iteration $t$. Then, $c^{(t)}(e)$ in 2-WL and 2-FWL tests is updated respectively by:

$$c^{(t)}(\boldsymbol{e}) = f\Big(c^{(t-1)}(\boldsymbol{e}), \{\!\{c^{(t-1)}(u, q)|u \in [n]\}\!\}, \{\!\{c^{(t-1)}(p, v)|v \in [n]\}\!\}\Big), \tag{3}$$

$$c^{(t)}(\boldsymbol{e}) = f_F\Big(c^{(t-1)}(\boldsymbol{e}), \{\!\{\big(c^{(t-1)}(u, q), c^{(t-1)}(p, u)\big)|u \in [n]\}\!\}\Big), \tag{4}$$

where $f, f^F$ are injective functions. For unlabeled graphs, we can take $c^{(0)}(e)$ to be the indicator of whether $e$ exists in $E$. For labeled graphs, we additionally consider the initial node features.

When the initial representation for link $(p, q)$ is its edge indicator, 2-FWL can **count the common neighbors** between $p, q$ by checking how many $(1, 1)$ appear in the multiset. By iterating on the third node $u$, it can actually learn all 3-node structures containing $p, q$.

Considering the space complexity and the locality of link prediction problems, we also propose local version of 2-WL and 2-FWL tests:

$$c^{(t)}(\boldsymbol{e}) = f^L\Big(c^{(t-1)}(\boldsymbol{e}), \{\!\{c^{(t-1)}(u, q)|(u, q) \in E\}\!\}, \{\!\{c^{(t-1)}(p, v)|(p, v) \in E\}\!\}\Big), \tag{5}$$

$$c^{(t)}(\boldsymbol{e}) = f_F^L\Big(c^{(t-1)}(\boldsymbol{e}), \{\!\{\big(c^{(t-1)}(u, q), c^{(t-1)}(p, u)\big)|(u, q) \in E \text{ or } (p, u) \in E\}\!\}\Big), \tag{6}$$

Figure **??** shows how local 2-FWL works. In brief, local version of tests only include neighbouring links that are connected. It's formally different from the local $k$-WL proposed in (Morris et al., 2019)

Note that such link-level WL test is different from graph-level tests as it directly output color of target units without pooling all units together. Though there has been results of $k$-WL power on graph isomorphism (Cai et al., 1992), The discussion of their link-level expressive power is still missing.

To compare the link discriminating power of 1-WL and different 2-WL variants, we first formally define 1-*WL-indistinguishable* and 2-*WL-indistinguishable*.

**Definition 2.1.** (1-*WL-indistinguishable*) Let $G = (V, E, l)$, $G' = (V', E', l')$ be two graphs, and $\boldsymbol{s} = (s_1, s_2, ..., s_k)$, $\boldsymbol{s}' = (s'_1, s'_2, ..., s'_k)$ be two equally sized node tuples, where $s_j \in V$, $s'_j \in V'$, $\forall j \in [k]$. Let $c^{(t)}(i)$ denote the color of node $i$ after $t$ steps of 1-WL update. If

$$(s_i, s_j) \in E \iff (s'_i, s'_j) \in E', \quad \forall i, j \in [k], \text{ and} \tag{7}$$

$$c^{(t)}(s_j) = c^{(t)}(s'_j), \quad \forall j \in [k], \forall t \geq 0, \tag{8}$$

we say $(\boldsymbol{s}, G)$ is 1-*WL-indistinguishable* from $(\boldsymbol{s}', G')$, denoted by $(\boldsymbol{s}, G) \simeq_{1\text{-}WL} (\boldsymbol{s}', G')$.

Note that we are more concerned with the case $G = G'$, $|\boldsymbol{s}| = |\boldsymbol{s}'| = 2$, where we aim to discriminate node pairs (links) on the same graph.

**Definition 2.2.** *(2-WL-indistinguishable) Given graphs* $G = (V, E, l)$, $G' = (V', E', l')$ *and links* $\boldsymbol{e} = (p, q) \in V^2$, $\boldsymbol{e}' = (p', q') \in V'^2$, $c^{(t)}(\boldsymbol{e})$ *the color of* $\boldsymbol{e}$ *after* $t$ *steps of* 2-WL *update, if*

$$c^{(t)}(\boldsymbol{e}) = c^{(t)}(\boldsymbol{e}'), \ \forall t \geq 0, \tag{9}$$

*we say* $(\boldsymbol{e}, G)$ *is* 2-WL-indistinguishable *from* $(\boldsymbol{e}', G')$, *denoted by* $(\boldsymbol{e}, G) \simeq_{\text{2-WL}} (\boldsymbol{e}', G')$.

Similarly, we can define indistinguishable property for other 2-WL variants that take links as update units. Note that for 2-WL-indistinguishable, we only consider the link case, but it can be generalized to arbitrary node tuples. Given Definition 2.1 and Definition 2.2, it is possible to compare the link discriminating power between 1-WL and 2-WL tests. Below we formally define the relative link discriminating power.

**Definition 2.3.** *(Discriminating Power) Given two tests* $\mathscr{A}$ *and* $\mathscr{B}$, *if* $\mathscr{A}$ *distinguishes* $(\boldsymbol{e}, G)$ *and* $(\boldsymbol{e}', G')$ ***only if*** $\mathscr{B}$ *distinguishes* $(\boldsymbol{e}, G)$ *and* $(\boldsymbol{e}', G')$ *for any* $\boldsymbol{e}, \boldsymbol{e}', G, G'$, *and there exists some* $\boldsymbol{e}_1, \boldsymbol{e}_1', G_1, G_1'$ *such that* $(\boldsymbol{e}_1, G_1)$ *is distinguishable from* $(\boldsymbol{e}_1', G_1')$ *by* $\mathscr{B}$ *but not by* $\mathscr{A}$, *then we say test* $\mathscr{B}$ *has* ***stronger*** *link discriminating power than test* $\mathscr{A}$, *denoted by* $\mathscr{A} \prec \mathscr{B}$. *If* $\mathscr{A}$ *distinguishes* $(\boldsymbol{e}, G)$ *and* $(\boldsymbol{e}', G')$ ***if and only if*** $\mathscr{B}$ *distinguishes* $(\boldsymbol{e}, G)$ *and* $(\boldsymbol{e}', G')$ *for any* $\boldsymbol{e}, \boldsymbol{e}', G, G'$, *we say test* $\mathscr{A}$ *has* ***equivalent*** *link discriminating power to test* $\mathscr{B}$, *denoted by* $\mathscr{A} \sim \mathscr{B}$.

The set of all link-level tests becomes a partially ordered set with relation $\prec$. Now we are able to compare the link expressive power between 1-WL and 2-WL variants.

## 3 THE POWER OF 2-WL TESTS FOR LINK PREDICTION

In this section we theoretically characterize the link discriminating power of different 2-WL tests by comparing them with each other and 1-WL. We summarize our results in Table 1.

### 3.1 2-WL AND 2-FWL TESTS HAVE STRONGER LINK DISCRIMINATING POWER THAN 1-WL

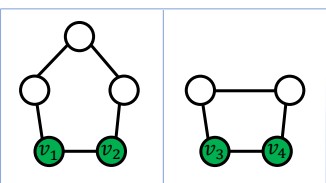

We use 2-WL to specifically denote its plain version defined in (3), and 2-FWL to denote the Folklore version defined in (4). We have the following theorems.

**Theorem 3.1.** *2-WL has stronger link discriminating power than 1-WL.*

**Theorem 3.2.** *2-FWL has stronger link discriminating power than 2-WL.*

Figure 3: Non-isomorphic links $(v_1, v_2)$ and $(v_3, v_4)$ from their respective graphs can be discriminated by 2-WL but not by 1-WL. 2-WL can capture global features like graph size but 1-WL only captures local structures.

Theorem 3.1 is proved by Theorem 3.3 and the example displayed in Figure 3. The proof of Theorem 3.2 is included in appendix. The two theorems directly derive that 1-WL$\prec$ 2-WL$\prec$ 2-FWL. 2-WL is strictly stronger than 1-WL on the link level because it still maintains the ability to capture global structure like unconnected component and the number of nodes rather than 1-WL. However, 2-WL still cannot discriminate links like $(v_1, v_2)$ and $(v_1, v_3)$ in Figure 1 or count common neighbors as the two branches of neighboring links $\{\!\!\{(u, q)|u \in [n]\}\!\!\}, \{\!\!\{(p, v)|v \in [n]\}\!\!\}$ from $(p, q)$ are still independently aggregated. 2-FWL integrates links $(u, p)$ and $(p, v)$ as $(u, v)$'s neighbours, therefore it has stronger link expressive power.

### 3.2 THE LINK DISCRIMINATING POWER OF LOCAL 2-WL AND LOCAL 2-FWL

2-WL and 2-FWL have higher link discriminating power than 1-WL. However, they also bring higher time and space complexity. Given a graph $G = (V, E)$ where $|V| = n$ and $|E| = m$, 1-WL takes $O(m)$ time complexity in each iteration and occupies $O(n)$ memory. For 2-WL, it requires $O(n^2)$ memory and $O(n^3)$ time for each iteration, which is unaffordable for large-scale graphs.

The local version of 2-WL and 2-FWL test denoted by 2-WL$_L$ and 2-FWL$_L$ relieve this question. They reduce the neighborhood scope from global to local thereby leveraging the graph sparsity and reducing the complexity. The following theorem characterizes local 2-WL's discriminating power.

**Theorem 3.3.** *2-WL$_L$ has equivalent link discriminating power to* 1-*WL.*

The whole proof is included in the appendix. The main idea is to establish an bijective mapping between the subtrees of 1-WL and those of 2-WL$_L$. Intuitively, the two neighborhoods of $e = (p, q)$ exactly correspond to the neighborhood of $q$ and $p$ in 1-WL.

Now we characterize the expressive power of 2-FWL$_L$.

**Theorem 3.4.** *2-FWL$_L$ has stronger link discriminating power than* 2-*WL$_L$ but lower than* 2-*FWL.*

The proof is included in the appendix. The rationale is similar that 2-FWL$_L$ test groups neighbor links by the shared nodes $u$, thus capturing higher-order information than 2-WL$_L$, and that 2-FWL$_L$ integrates only local information than 2-FWL.

Summarizing previous results, we depict a full picture of the relative link discriminating power of all the tests in Table 1. In general, the original 2-WL tests are stronger than their local versions, and the Folklore versions are stronger than the plain versions. All 2-WL tests except the 2-WL$_L$ are stronger than 1-WL. Although the local versions are less powerful, they bring significant complexity reduction, as well as possibly more robustness and better generalizability for link prediction due to their focus on local structure patterns. Our experiments verify that local versions are usually not worse.

Table 1: The below matrix shows relative link discriminating power of different tests, where $\sim$ denotes equal power, $\prec$ denotes weaker power, and - denotes that both are not weaker than the other.

|            | 1-WL | 2-WL$_L$ | 2-WL | 2-FWL$_L$ | 2-FWL |
|------------|------|----------|------|-----------|-------|
| 1-WL       | $\sim$ | $\sim$ | $\prec$ | $\prec$ | $\prec$ |
| 2-WL$_L$   |      | $\sim$ | $\prec$ | $\prec$ | $\prec$ |
| 2-WL       |      |      | $\sim$ | - | $\prec$ |
| 2-FWL$_L$  |      |      |      | $\sim$ | $\prec$ |
| 2-FWL      |      |      |      |      | $\sim$ |

## 4 IMPLEMENTATION BY GNN MODELS

We first use 1-WL-GNN to learn node embeddings with the raw node features inspired by (Morris et al., 2019). If raw node features don't exist, we take embeddings of node degrees to keep the inductive property of our model. Then, we obtain the initial link representations by pooling the pairwise node embeddings. Then we implement four link-level 2-WL tests through GNNs in a totally different way due to different neighborhood aggregation form and goal of complexity reduction.

### 4.1 GNN IMPLEMENTATION OF 2-WL

Constructing a complete graph and apply traditional graph convolutions suffers from $O(n^3)$ time complexity is unaffordable for large graphs. Therefore, we construct our own aggregation and combination functions. We group the link representations in the $t^{\text{th}}$ step into an $n \times n \times r$ tensor $A^{(t)}$, where the $p, q$ indexed vector $A_{p,q,:}^{(t)}$ is the representation of link $(p, q)$. For $A^{(0)}$, we also include the adjacency matrix as one slice. Then $A^{(t+1)}$ is computed by:

$$B_{p,q,:}^{(t)} = concat\big(\sum_{i \in [n]} g(A_{p,i,:}^{(t)}), \sum_{i \in [n]} h(A_{i,q,:}^{(t)})\big), \quad \text{(Aggregation)} \tag{10}$$

$$A^{(t+1)} = f\big(concat(B^{(t)}, A^{(t)})\big), \quad \text{(Combination)} \tag{11}$$

where $f, g, h$ are MLPs. Given the ordered node pair $(p, q)$, in each layer we apply two distinct transformations $g$ and $h$ to respectively aggregate its neighbors $\{\!\{(u, q) | u \in [n]\}\!\}$, $\{\!\{(p, v) | v \in [n]\}\!\}$. Directly operating on the dense link representations $A$ saves us from explicitly constructing the complete graph, and allows using MLP to implement the model. The whole algorithm takes $O(n^2 r^2)$ time and $O(n^2 r)$ memory.

### 4.2 GNN IMPLEMENTATION OF 2-WL$_L$

Local 2-WL is realised differently from 2-WL. Due to the reduced neighborhood, it is possible to leverage the graph sparsity to save memory and time. In training period, let $S$ be the mini-batch containing all positive and negative target links to predict, $E'$ be the existing edges in the original graph (after removing the positive training links). Then we construct a second-order graph

$G_S := (E' \cup S, E^{(2)})$, where $E^{(2)}$ denotes "edges" between $E' \cup S$ based on the neighborhood definition of 2-WL$_L$. We then apply graph convolution on $G_S$ to obtain node representations for $S$ which are used to output their link prediction scores in the original graph. The second-order graph has $O((|E'| + |S|)d)$ edges, where $d$ is the max node degree in the original graph. Therefore, the time complexity of message passing follows to be $O((|E'| + |S|)dr^2)$, where $r$ is the hidden dimension. Memory efficiency is also largely improved to $O((|E'| + |S|)r)$.

## 4.3 GNN IMPLEMENTATION OF 2-FWL

The situation becomes a bit more complex for 2-FWL. The join of two links is difficult to implement by standard graph convolution layers. Thus, we apply a model similar to that proposed in Maron et al. (2019). In each layer, we apply slice-wise matrix multiplication of two reshaped link representation tensors to implement the 2-FWL message passing.

$$B_{p,q,:}^{(t)} = \sum_{i \in [n]} g(A_{p,i,:}^{(t)}) \odot h(A_{i,q,:}^{(t)}), \quad \text{(Aggregation)} \tag{12}$$

$$A^{(t+1)} = f\big(concat(B^{(t)}, A^{(t)})\big), \quad \text{(Combination)} \tag{13}$$

where $\odot$ is element-wise product and $f, g, h$ are MLPs. The above implementation first joins link representations of $(p, i)$ and $(i, q)$ through element-wise product, and then performs the aggregation through summing. We also add adjacency matrix into the first layer.

Because 2-FWL uses batched matrix multiplication, it has time complexity $O(n^3r + n^2r^2)$ and space complexity $O(n^2r)$, with $r$ the third dimension of $A^{(t)}$. It seems to be slower than 2-WL model, but batched matrix multiplication can be efficiently computed in practice.

## 4.4 GNN IMPLEMENTATION OF 2-FWL$_L$

For local 2-FWL, the implementation is the same as 2-FWL except that we replace the dense matrix multiplications in Equation (12) with sparse matrix multiplications, i.e., initially only those entries $A_{p,q,:}^{(0)}$ corresponding to existing edges $(p, q) \in E$ have nonzero values, and at the $t^{\text{th}}$ message passing step we still only track those nonzero $p, q$ entries. We also concatenate $n \times n \times 1$ adjacency matrix to embedding in each layer.

Note that this implementation doesn't perfectly follow the definition of 2-FWL$_L$ in (6) and therefore cannot learn representations for all (intermediate) links (which may fails to cover questioned links). Thus, we concatenate the final link representations with node-pair representations learned by a 1-WL-GNN to give a nonzero representation to any questioned link. Although this implementation does not preserve the full representation power of 2-FWL$_L$, it can still learn common neighbor and path-counting features between nodes, and most importantly, it significantly reduces the space complexity from $O(n^2r)$ to $O(md^lr)$, with $d$ max node degree, $l$ number of layers, $r$ hidden dimension.

## 5 RELATED WORK

Weisfeiler-Lehman tests are a family of algorithms to deal with the graph isomorphism problem (Cai et al., 1992). In addition to graph isomorphism checking, they have found many applications in machine learning recently (Morris et al., 2021). Shervashidze et al. (2011) use the idea to construct subtree-based graph kernels. Niepert et al. (2016) and (Zhang & Chen, 2017b) use WL to sort nodes and construct neural networks for graphs. Vanilla GNNs have also been shown to have limited graph discriminating power bounded by 1-WL (Xu et al., 2019). Many works focus on how to improve GNNs' power by considering high-dimensional WL tests. Morris et al. (2019) introduce GNN models simulating 2-WL and 3-WL tests. Maron et al. (2019); Chen et al. (2019b) achieve the same graph discriminating power as 3-WL with a 2-FWL based model. However, these works all deal with the whole-graph representation learning problem. Little work has been done in the link prediction context. The graph-level tasks mainly work on small graph when encoding high-order graph structure, but graphs for link prediction can be extremely large and more attention should be put on their local property. Therefore whether the 2-WL based models work well in link-level tasks and how to design their variants to adapt to the locality are valuable questions. In this work, we also for the first

time demonstrate both the theoretical and practical power of 2-WL-based GNNs for link prediction, therefore filling in this blank area.

In the community of using GNN models for link-oriented tasks, various techniques have been proposed to enhance their theoretical power. SEAL (Zhang & Chen, 2018) utilizes a distance-based node labeling trick to label the context nodes according to their relationships to the target link, which is later formalized into distance encoding (Li et al., 2020). Zhang et al. (2021) further proved that such a labeling trick brings theoretical improvement to GNNs' link discriminating power. However, using labeling tricks requires extracting a subgraph for each link and repeatedly applying GNN to the subgraphs, which incurs high computational complexity and prevents full-batch learning. In contrast, our models aim to still apply GNN only once to the entire graph like the traditional GAE methods, while outperforming GAE in terms of link discriminating power. NBFNet (Zhu et al., 2021) uses a type of partial labeling trick which only labels the source node and applies a GNN to predict all links from the source node. Although it does not need to extract a subgraph for every link, it needs to apply a GNN to a large graph for each source node and suffers from low training efficiency. On the basis of SEAL, Pan et al. (2022) encode a transition matrix serving as a form of pairwise encoding for each link in the subgraph. However, it still requires extracting subgraphs for all links to predict.

Given original graph $G$, line graph $L(G)$ represents the adjacency between edges. In $L(G)$, each node corresponds to a unique edge in $G$. By using node representation learning methods (Kipf & Welling, 2017) on the line graph, some methods (Zhu et al., 2019; Chen et al., 2019a; Jiang et al., 2019; Cai et al., 2021; Liu et al., 2021) can utilize edge features and topology better, which have achieved outstanding performance on graph tasks like heterogeneous graph learning, community detection, graph classification, and link prediction. Using 1-WL-GNNs on line graphs is similar to local 2-WL. However, none of these previous works have noticed the connection between line graph and 2-WL tests. Furthermore, more expressive variants like 2-FWL are not studied.

## 6  EXPERIMENTS

In this section, we conduct experiments to verify the effectiveness of 2-WL-GNNs for link prediction. We test 2-WL-GNNs based on the proposed four tests: 2-WL, local 2-WL (2-WL$_L$), 2-FWL, and local 2-FWL (2-FWL$_L$). Since these methods do message passing among two-node tuples rather than sets, they can do prediction tasks on both homogeneous graph data and knowledge graph data. To predict the edge between nodes $p, q$ in former condition, we simply pool $(p, q)$ and $(q, p)$'s embedding to obtain $\{p, q\}$'s embedding as the prediction score. Hyperparameters include learning rate, hidden dimension, number of message passing layers, and dropout rate. Baseline results are taken from (Zhang & Chen, 2018), (Liu et al., 2021) and (Zhu et al., 2021).

### 6.1  HOMOGENEOUS GRAPH LINK PREDICTION

The baseline methods we choose are Matrix Factorization (MF) (Mnih & Salakhutdinov, 2008), Node2Vec (N2V) (Grover & Leskovec, 2016), Weisfeiler-Lehman Neural Machine (WLNM) (Zhang & Chen, 2017a), TLC-GNN (Yan et al., 2021), 1-WL-GNNs including VGAE (Kipf & Welling, 2016) and S-VGAE (Davidson et al., 2018), and labeling trick methods including SEAL (Zhang & Chen, 2018), NBFNet (Zhu et al., 2021), WalkPool (Pan et al., 2022) and Neo-GNN (Yun et al., 2021). We use eleven benchmark datasets. Three of them are citation networks with node feature information: Cora, CiteSeer and Pubmed (Sen et al., 2008). The other eight datasets are: USAir, NS, PB, Yeast, C.ele, Power, Router, and E.coli from SEAL, which are networks from different domains and do not contain node features. For each network, we randomly choose $10\%$ edges as test set and $5\%$ edges as validation set. The remaining are treated as the observed training graph. The same number of randomly sampled nonexistent links are added into each set as the negative data. The performance metric is area under the ROC curve (AUC). We run each model for 10 times and report the average performance and standard deviations.The results are presented in Table 2 and 3.

According to the results, our 2-WL-GNNs achieve generally better performance than the baseline models. Specifically, the 2-WL and 2-WL$_L$ models perform competitively with SEAL on a large number of datasets and the 2-FWL and 2-FWL$_L$ models obtain overall better results than SEAL. On citation networks our models also achieve competitive results among recent methods. As displayed in Table 3, 2-FWL achieves a new state-of-the-art result of 96.44 on Citeseer. On Cora and Pubmed,

Table 2: Performance on eight networks without node features

| Dataset | MF | N2V | VGAE | WLNM | SEAL | 2-WL | 2-WL$_L$ | 2-FWL | 2-FWL$_L$ |
|---|---|---|---|---|---|---|---|---|---|
| USAir | 94.08±0.80 | 91.44±1.78 | 89.28±1.99 | 95.95±1.10 | 97.09±0.70 | 92.86± 1.08 | 94.65± 0.99 | **98.10± 0.42** | 96.06± 0.51 |
| NS | 74.55±4.34 | 91.52±1.28 | 94.04±1.64 | 98.61±0.49 | 97.71±0.93 | 97.15± 0.78 | 95.79± 0.73 | 98.85± 0.43 | **99.49± 0.12** |
| PB | 94.30±0.53 | 85.79±0.78 | 90.70±0.53 | 93.49±0.47 | 95.01±0.34 | 93.61± 0.54 | **95.10± 0.48** | 94.07± 0.47 | 94.71± 0.54 |
| Yeast | 90.28±0.69 | 93.67±0.46 | 93.88±0.21 | 95.62±0.52 | 97.20±0.64 | 95.76± 0.54 | 95.33± 3.34 | **97.82± 0.21** | 97.44± 0.25 |
| Cele | 85.90±1.74 | 84.11±1.27 | 81.80±2.18 | 86.18±1.72 | 86.54±2.04 | 81.72± 2.15 | 83.34± 2.35 | **91.25± 3.82** | 88.68± 1.34 |
| Power | 50.63±1.10 | 76.22±0.92 | 71.20± 1.65 | 84.76±0.98 | 84.18±1.82 | 74.10± 1.90 | 81.02± 1.25 | 72.21± 1.16 | **85.60±0.67** |
| Router | 78.03±1.63 | 65.46±0.86 | 61.51±1.22 | 94.41±0.88 | 95.68±1.22 | 96.02± 0.61 | **96.68± 0.50** | 95.34± 0.79 | 94.91±0.64 |
| Ecoli | 93.76±0.56 | 90.82±1.49 | 90.81±0.63 | 97.21±0.27 | 97.22±0.28 | 96.12± 0.48 | 96.51± 0.28 | **98.42± 0.21** | 97.03± 0.57 |

Table 3: Performance on citation networks with node features. OOM: Out of memory.

| Dataset | VGAE | S-VGAE | TLC-GNN | SEAL | NBFNet | WalkPool | Neo-GNN | 2-WL | 2-WL$_L$ | 2-FWL | 2-FWL$_L$ |
|---|---|---|---|---|---|---|---|---|---|---|---|
| Cora | 91.4 | 94.1 | 93.4 | 93.3 | 95.6 | 96.0 | **96.2** | 93.15±1.17 | 95.33±0.50 | 96.03±0.52 | 95.81±0.60 |
| Citeseer | 90.8 | 94.7 | 90.9 | 90.5 | 92.3 | 96.0 | 95.5 | 94.45±1.01 | 92.10± 0.79 | 95.28± 0.76 | **96.44±0.67** |
| Pubmed | 94.4 | 96.0 | 97.0 | 97.8 | 98.3 | 98.6 | **99.2** | OOM | 98.66± 0.16 | OOM | 98.69±0.09 |

2-FWL and 2-FWL$_L$ also obtain second best results respectively. Their competitive performance verifies the effectiveness of directly using links as message passing units to learn their representations.

Theoretically, both labeling trick methods and 2-FWL models are more expressive than 1-WL models like VGAE and S-VGAE, which is reflected in their performance comparisons. However, we found even 2-WL and local 2-WL models can sometimes outperform 1-WL-GNNs by large margins, especially on networks without node features. This might be explained by that the direct learning of link representations and the message passing along edge adjacency might capture better edge topology than node-centered methods. Furthermore, we found that the global versions 2-WL and 2-FWL do not always achieve better performance than their local versions 2-WL$_L$ and 2-FWL$_L$, despite being theoretically more powerful. This might be because the local versions focus more on local neighborhood around links, which is proved to contain the most useful information for link prediction (Zhang & Chen, 2018). Considering the significantly larger memory requirement (OOM in Pubmed), we recommend to use the local versions in most cases due to their efficiency and scalability.

## 6.2 KNOWLEDGE GRAPH COMPLETION

we conduct an additional experiments to test 2-WL-GNNs' link prediction performance on inductive knowledge graph completion (KGC). We adopt two datasets, FB15K-237 and WN18RR from (Teru et al., 2020) to evaluate the performance. Each dataset includes four versions v1 to v4 with increasing sizes. The baselines we use are state-of-the-art inductive KGC methods including R-GCN (Schlichtkrull et al., 2017), GraIL (Teru et al., 2020), and a recent line-graph-based model INDIGO (Liu et al., 2021). We compare them with our GNN implementations of 2-WL$_L$ and 2-FWL$_L$ using three metrics: accuracy (ACC), area under the ROC curve (AUROC) and Hits@3. The results are given in Table 4. Best and second-to-best results are in bold and with underlines respectively.

As we can see, 2-FWL$_L$ generally achieves the strongest performance with **17 highest metric numbers out of 24**, and 3 second-to-best metric numbers among the remaining. This again verifies the higher link expressive power brought by the 2-FWL tests. On the other hand, 2-WL$_L$ and INDIGO perform competitively too. As discussed in the related work, INDIGO can be understood as a special implementation of local 2-WL by leveraging line graphs. The excellent performance of 2-WL methods further verifies the advantage of directly learning link representations. Specifically, we notice that these link-centered methods have much higher Hits@3 than the other node-centered baselines, indicating that link-centered methods are better at ranking the correct links at the top. This is especially important in real-world applications where we can only focus on top-ranked predictions.

## 6.3 TIME COMPLEXITY ANALYSIS

Finally, we compare time complexity between local 2-WL models and labeling trick methods. For each model, we record the time to train an epoch as training, and the computation of prediction scores for all links of test set as the inference time. We use the default batch size: 10% of the number of all existed links in 2-WL models, 32 in SEAL (dynamic train/test), $(64, 64, 16)$ for Cora, Citeseer, Pubmed respectively in NBFNet. Table 5 summarize the results. Best and second-to-best results are in bold and with underline, respectively. The performance demonstrates that 2-WL based models

Table 4: Performance on KG datasets (%). Higher the better.

| | | FB15K-237 | | | | WN18RR | | | |
|---|---|---|---|---|---|---|---|---|---|
| | | v1 | v2 | v3 | v4 | v1 | v2 | v3 | v4 |
| ACC | R-GCN | 51.0 | 51.3 | 54.9 | 52.1 | 50.2 | 52.7 | 52.2 | 48.4 |
| | GraIL | 69.0 | 80.0 | 81.0 | 79.3 | **88.7** | 81.2 | 75.7 | 86.4 |
| | INDIGO | 84.3 | 89.3 | 89.0 | 87.8 | 85.7 | 85.8 | **84.3** | 85.4 |
| | 2-WL$_L$ | 85.7 | 93.2 | 90.0 | 91.1 | 84.7 | 86.5 | 79.9 | 86.8 |
| | 2-FWL$_L$ | **90.7** | **94.7** | **93.9** | **91.8** | 84.7 | **86.7** | 81.5 | **88.7** |
| AUROC | R-GCN | 51.0 | 50.5 | 50.5 | 52.6 | 49.0 | 49.8 | 53.1 | 50.2 |
| | GraIL | 78.6 | 90.0 | 93.1 | 89.5 | 92.3 | 92.7 | 82.8 | 94.4 |
| | INDIGO | 93.4 | 96.3 | 96.6 | 95.8 | 91.2 | 92.5 | **92.4** | 94.7 |
| | 2-WL$_L$ | 87.9 | 95.7 | 96.9 | **97.7** | 88.5 | 93.3 | 86.6 | 89.1 |
| | 2-FWL$_L$ | **95.3** | **98.2** | 97.5 | 96.6 | 92.8 | 93.3 | 85.9 | **95.4** |
| Hits@3 | R-GCN | 2.4 | 3.4 | 3.5 | 3.3 | 2.1 | 11.0 | 24.5 | 8.1 |
| | GraIL | 1.0 | 0.4 | 6.6 | 3.0 | 0.6 | 10.7 | 17.5 | 22.6 |
| | INDIGO | 53.1 | 67.6 | 66.5 | 66.3 | **98.4** | **97.3** | **91.9** | 96.1 |
| | 2-WL$_L$ | 70.8 | 79.0 | 79.5 | 79.8 | 97.8 | 96.1 | 83.7 | 96.2 |
| | 2-FWL$_L$ | **71.5** | **84.2** | **81.7** | 78.3 | 97.4 | 96.6 | 85.2 | **97.3** |

Table 5: Time complexity comparison.

| Dataset | Training Time | | | | | | Inference Time | | | | | |
|---|---|---|---|---|---|---|---|---|---|---|---|---|
| | 2-WL$_L$ | 2-WL | 2-FWL$_L$ | 2-FWL | SEAL | NBFNet | 2-WL$_L$ | 2-WL | 2-FWL$_L$ | 2-FWL | SEAL | NBFNet |
| Cora | **0.54**s | 4.59s | 9.18s | 6.03s | 8.44s | 48.38s | **0.007**s | 0.29s | 1.45s | 0.16s | 2.30s | 1.94s |
| Citeseer | **0.90**s | 4.32s | 5.85s | 4.41s | 7.69s | 49.15s | **0.006**s | 0.59s | 0.74s | 0.24s | 2.11s | 1.80s |
| Pubmed | **13.05**s | OOM | 38.43s | OOM | 59.67s | 3000s | **0.05**s | OOM | 3.9s | OOM | 15.4s | 95s |

have significantly lower time complexity than labeling trick methods. This is because local 2-WL models can predict all the target links by applying the GNN once to the entire graph, while labeling trick methods require repeatedly applying GNNs to a labeled graph for every target link or source node to predict. 2-FWL$_L$ is relatively slower due to inefficient sparse matrix operation, but it brings space efficiency.

### 6.4 LIMITATIONS

A notable problem for 2-WL models is its scalability. Although we purpose local version of the plain 2-WL models, they still suffers from high space complexity and relatively low performance when dealing with large and dense dataset. On Power and Router with nearly 5000 nodes, our models perform not much better than baselines. On Pubmed with 19717 nodes and 44324 edges, 2-WL and 2-FWL used up 48GB memories and their local versions fail to outperform Neo-GNN. All of our models cannot work on OGBL dataset due to the large size. One possible solution is using graph partition. Despite such weakness, 2-WL models obtain remarkable performance on small graph and speed advantages.

## 7 CONCLUSIONS

This work studies a novel method for link prediction based on two-dimensional Weisfeiler-Lehman graph neural networks. The problems with the prevalent 1-WL based models is long-standing. Even though node labeling fundamentally improve the theoretical power, it brings low efficiency in training and inference process. 2-WL-GNN can resolve both challenges. Free of any labeling, it doesn't break the symmetry of the whole graph thereby having a unique speed advantage. Meanwhile, we theoretically characterize the link discriminating power of different 2-WL variants, and show that except local 2-WL, all other tests have stronger power than 1-WL. Besides, we further construct a series of GNNs implementing the 2-WL tests. They follows our proposed 2-WL's patterns and successfully exploit their expressive power. Experiments on multiple benchmark datasets show the effectiveness of 2-WL-GNNs for link prediction. Despite scalability problems for large dataset, our results shows their impressive performance on small dataset in different tasks. More importantly, we present a novel paradigm for link prediction with distinctive advantages, producing new direction to discover.

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

## A    PROOF OF THEOREM 3.3

**Theorem**: 2-WL$_L$ has the same discriminating power as 1-WL for link prediction.

*Proof.* We measure the link discriminating power by constructing subtrees. Given an undirected graph $G = (V, E, l)$, $p, q \in V$, denote edge indicator of $(p, q)$ as $1_{(p,q)}$ and let $T_{\mathscr{A}}, T_{\mathscr{B}}$ be mappings from sets of graph-link tuples $\big(G, (p, q)\big)$ to sets of tree-structured graphs with infinite depth, which are defined as follows.

For graph $G$, $p, q \in G$, $|G| = n$. $T_{\mathscr{A}}\big(G, (p, q)\big)$ has a root $(p, q)$ labeled as $\big(l(p), l(q), 1_{(p,q)}\big)$ with two branches of child nodes $\{(p, i) : (p, i) \in E, i \in [n]\}$ and $\{(j, q) : (j, q) \in E, j \in [n]\}$ in the left and right side, respectively. For every child node $(r, s)$, it is labeled as $\big(l(r), l(s)\big)$. Its child nodes and their labels are defined in the same way recursively.

$T_{\mathscr{B}}\big(G, (p, q)\big)$ has a root $(p, q, 1_{(p,q)})$ which is labeled as $\big(l(p), l(q)\big)$. It has two branches of child nodes: $\{i : (p, i) \in E, i \in [n]\}$ on the left and $\{j : (j, q) \in E, j \in [n]\}$ on the right. In the following layers node $k$ has children $\{l : (k, l) \in E, l \in [n]\}$. Node $k$ is labeled in the graph as $l(k)$.

Then we define an equivalent class across the tree-structured graphs: Denote $E_T = \{(prec, next, br) : next$ is the child node of $prec$ in branch $br$ in tree $T\}$. If there is a bijective mapping $\pi$ from nodes of a finite-depth tree $T_1$ (denoted by $V(T_1)$) to nodes of a finite-depth tree $T_2$ (denoted by $V(T_2)$) such that 1) $l(i) = l\big(\pi(i)\big), \forall i \in V(T_1)$, 2) $(i, j, br) \in E_{T_1} \iff \big(\pi(i), \pi(j), br\big) \in E_{T_2}, \forall i, j \in V(T_1)$, we say $T_1$ is equivalent to $T_2$, denoted as $T_1 \simeq T_2$.

Let $T|_k$ refer to the mapping that $T|_k(G, e)$ is the first $k$ layers of subtree $T(G, e)$. We define that two infinite-depth trees $T_1, T_2$ satisfy $T_1 \simeq T_2$ if and only if $T_1|_k \simeq T_2|_k, \forall k \in \mathbb{N}$.

Given the well defined equivalent class and Definition 2.1, we notice that $T_{\mathscr{A}}, T_{\mathscr{B}}$ depict the process of local 2-WL and 1-WL test, that is,

$$\big((p, q), G\big) \simeq_{\text{2-WL}_L} \big((p', q'), G'\big) \iff T_{\mathscr{A}}\big(G, (p, q)\big) \simeq T_{\mathscr{A}}\big(G', (p', q')\big) \tag{14}$$

$$\big((p, q), G\big) \simeq_{\text{1-WL}} \big((p', q'), G'\big) \iff T_{\mathscr{B}}\big(G, (p, q)\big) \simeq T_{\mathscr{B}}\big(G', (p', q')\big) \tag{15}$$

Therefore the statement that local 2-WL and 1-WL has equivalent link discriminating power equals to that $\forall (G, e), (G', e')$,

$$T_{\mathscr{A}}(G, e) \simeq T_{\mathscr{A}}(G', e') \iff T_{\mathscr{B}}(G, e) \simeq T_{\mathscr{B}}(G', e'). \tag{16}$$

According to our definition, we need to prove that for $\forall k \in \mathbb{N}$,

$$T_{\mathscr{A}}|_k(G, e) \simeq T_{\mathscr{A}}|_k(G', e') \iff T_{\mathscr{B}}|_k(G, e) \simeq T_{\mathscr{B}}|_k(G', e'), \; \forall (G, e), (G', e') \tag{17}$$

For $k = 0$, we have $T_{\mathscr{A}}|_0(G, e) \simeq T_{\mathscr{A}}|_0(G', e') \iff l(p) = l(p'), \; l(q) = l(q'), \; 1_{(p,q)} = 1_{(p',q')} \iff T_{\mathscr{B}}|_0(G, e) \simeq T_{\mathscr{B}}|_0(G', e')$

Suppose (15) works for $k = L$, let's consider the situation of $k = L + 1$:

Denote $\{i_1, i_2, ..., i_{n_p}\}$, $\{j_1, j_2, ..., j_{n_q}\}$ as neighbors of $p, q$ in $G$, and $\{i'_1, i'_2, ..., i'_{n_{p'}}\}$, $\{j'_1, j'_2, ..., j'_{n_{q'}}\}$ as neighbors of $p', q'$ in $G'$, respectively. According to the property of local 2-WL test, if $T_{\mathscr{A}}|_{L+1}(G, e) \simeq T_{\mathscr{A}}|_{L+1}(G', e')$, we have $l(p) = l(p'), \; l(q) = l(q'), \; n_p = n_{p'}, \; n_q = n_{q'}$ and w.l.o.g.

$$T_{\mathscr{A}}|_L(G, (p, i_s)) \simeq T_{\mathscr{A}}|_L(G', (p', i'_s)), \; \forall s \in [n_p] \tag{18}$$

$$T_{\mathscr{A}}|_L(G, (j_t, q)) \simeq T_{\mathscr{A}}|_L(G', (j'_t, q')), \; \forall t \in [n_q] \tag{19}$$

Since (15) works for $k = L$, we have

$$T_{\mathscr{B}}|_L\big(G, (p, i_s)\big) \simeq T_{\mathscr{B}}|_L(G', (p', i'_s)), \; \forall s \in [n_p] \tag{20}$$

$$T_{\mathscr{B}}|_L\big(G, (j_t, q)\big) \simeq T_{\mathscr{B}}|_L(G', (j'_t, q')), \; \forall t \in [n_q] \tag{21}$$

According to property of 1-WL test, (18), (19) mean $T_{\mathscr{B}}|_{L+1}\big(G, (p, q)\big)$ and $T_{\mathscr{B}}|_{L+1}\big(G', (p', q')\big)$ have $m + n$ correspondingly isomorphic $L$-depth branches. Plus $l(p) = l(p'), \; l(q) = l(q')$ we conclude $T_{\mathscr{B}}|_{L+1}\big(G, (p, q)\big) \simeq T_{\mathscr{B}}|_{L+1}\big(G', (p', q')\big)$. The other direction can be similarly proved. Figure 4 gives an illustration. $\qquad \square$

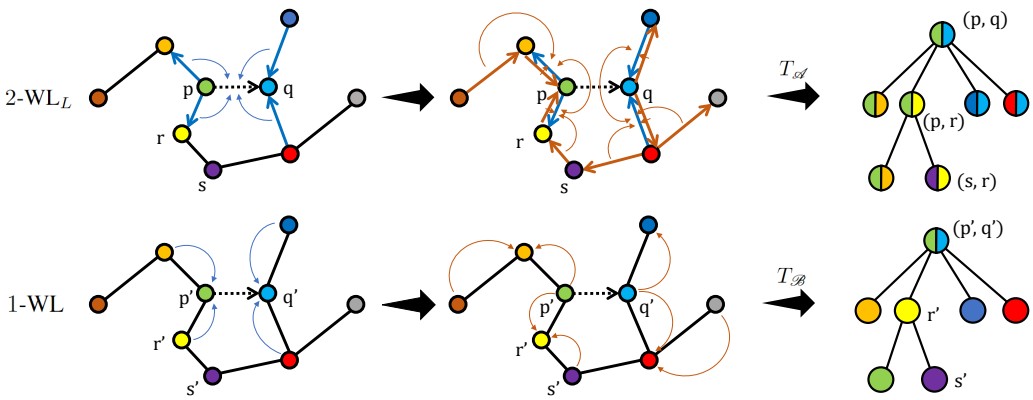

Figure 4: Update patterns of 2-WL$_L$ and 1-WL test, and their corresponding mappings from $(G, e)$ to subtrees in our proof. We can build a one-to-one mapping between subtrees of 2-WL$_L$ and 1-WL. We show only part of subtrees.

# B  PROOF OF THEOREM 3.2 AND THEOREM 3.4

**Theorem:** 2-FWL has stronger link discriminating power than 2-WL.

*Proof.* Let $T_{\mathscr{C}}, T_{\mathscr{D}}$ be mappings from sets of graph-link tuples to sets of tree-structured graphs with infinite depth.

For graph $G = (V, E, l)$, $p, q \in V$, $|G| = n$. $T_{\mathscr{C}}(G, (p, q))$ has a root $(p, q)$ labeled as $(l(p), l(q), 1_{(p,q)})$ with two branches of child nodes $\{(p, i) : i \in [n]\}$ and $\{(j, q) : j \in [n]\}$ on the left and right side, respectively. For every child node $(r, s)$, its child nodes are defined in the same way recursively. Node $(r, s)$ is labeled as $(l(r), l(s), 1_{(r,s)}, 1_{\{r=s\}})$.

$T_{\mathscr{D}}(G, (p, q))$ has root $(p, q)$ labeled as $(l(p), l(q), 1_{(p,q)})$. It has $n$ child nodes $\{((p, i), (i, q)) | i \in [n]\}$. Node $((p, r), (r, q))$ is labeled as $(l(p), l(r), l(q), 1_{\{(p,r) \in E\}}, 1_{(r,q)}, 1_{\{p=r\}}, 1_{\{r=q\}})$ which has two branches of child nodes $\{((p, t), (t, r)) | t \in [n]\}$ and $\{((r, s), (s, q)) | s \in [n]\}$. Each of them has its label and child nodes defined in the same way recursively.

Therefore $T_{\mathscr{C}}$ and $T_{\mathscr{D}}$ depict the process of 2-WL and 2-FWL tests. After defining the equivalent class of tree-structured graph as in the proof of Theorem 3.3, we have

$$(G, (p, q)) \simeq_{\text{2-WL}} (G', (p', q')) \iff T_{\mathscr{C}}(G, (p, q)) \simeq T_{\mathscr{C}}(G', (p', q')) \tag{22}$$

$$(G, (p, q)) \simeq_{\text{2-FWL}} (G', (p', q')) \iff T_{\mathscr{D}}(G, (p, q)) \simeq T_{\mathscr{D}}(G', (p', q')) \tag{23}$$

Let $T|_k$ refer to the mapping such that $T|_k(G, e)$ is the first $k$ layers of subtree $T(G, e)$. We will prove that for $\forall k \in \mathbb{N}$,

$$T_{\mathscr{D}}|_k(G, e) \simeq T_{\mathscr{D}}|_k(G', e') \Rightarrow T_{\mathscr{C}}|_k(G, e) \simeq T_{\mathscr{C}}|_k(G', e'), \ \forall (G, e), (G', e') \tag{24}$$

Fix $(G, e), (G', e'), G = (V, E, l), G' = (V', E', l')$. Let $n = |V|, n' = |V'|$. When $k = 0$, $T_{\mathscr{D}}|_0(G, e) \simeq T_{\mathscr{D}}|_0(G', e') \Rightarrow l(p) = l(p'), l(q) = l(q'), 1_{(p,q)} = 1_{(p',q')} \Rightarrow T_{\mathscr{C}}|_0(G, e) \simeq T_{\mathscr{C}}|_0(G', e')$

Suppose (24) is true for $k = L, L \geq 0$. Let's consider the situation of $k = L + 1$. According to the property of 2-FWL test, if $T_{\mathscr{D}}|_{L+1}(G, e) \simeq T_{\mathscr{D}}|_{L+1}(G', e')$, we immediately have $n = n'$ and

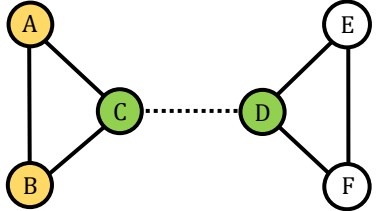 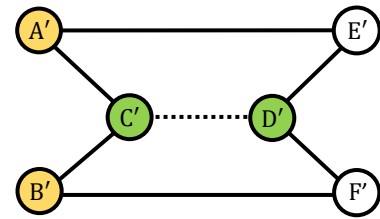

Figure 5: This figure contains two counterexample. First, links $(A, B)$ and $(A', E')$ cannot be distinguished by plain 2-WL but can be distinguished by (local) 2-FWL and 1-WL with 0/1 labeling trick. In fact due to high node-level symmetry 2-WL cannot detect difference between any connected link pairs or unconnected link pairs. The labeling trick breaks such symmetry and help 1-WL to capture the difference of two graph's structure. If $(C, D)$ and $(C', D')$ are target node pairs, 0/1 labeling trick no longer works. However, 2-FWL and local 2-FWL still work (because $(D, E)$ and $(D', E')$ will have different representations). They can capture triple structure as 3-WL test does.

w.l.o.g.

$$l(i) = l(i'), \ \forall i \in [n] \tag{25}$$

$$1_{(p,i)} = 1_{(p',i')}, \ \forall i \in [n] \tag{26}$$

$$1_{(i,q)} = 1_{(i',q')}, \ \forall i \in [n] \tag{27}$$

$$\Big( T_{\mathscr{D}}|_L\big(G, (p,i)\big), T_{\mathscr{D}}|_L\big(G, (i,q)\big) \Big) \simeq \Big( T_{\mathscr{D}}|_L\big(G', (p',i')\big), T_{\mathscr{D}}|_L\big(G', (i',q')\big) \Big), \ \forall i \in [n] \tag{28}$$

Then we have

$$T_{\mathscr{D}}|_L\big(G, (p,i)\big) \simeq T_{\mathscr{D}}|_L\big(G', (p',i')\big), \ \forall i \in [n] \tag{29}$$

$$T_{\mathscr{D}}|_L\big(G, (j,q)\big) \simeq T_{\mathscr{D}}|_L\big(G', (j',q')\big), \ \forall j \in [n] \tag{30}$$

Due to that (24) is true for $k = L$, we have

$$T_{\mathscr{C}}|_L\big(G, (p,i)\big) \simeq T_{\mathscr{C}}|_L\big(G', (p',i')\big), \ \forall i \in [n] \tag{31}$$

$$T_{\mathscr{C}}|_L\big(G, (j,q)\big) \simeq T_{\mathscr{C}}|_L\big(G', (j',q')\big), \ \forall j \in [n] \tag{32}$$

According to (23), (24), (25), (29), (30) and the definition of $T_{\mathscr{C}}$, we have

$$T_{\mathscr{C}}|_{L+1}\big(G, (p,q)\big) \simeq T_{\mathscr{C}}|_{L+1}\big(G', (p',q')\big), \ \forall i \in [n] \tag{33}$$

On the other side the counterexample lies in Figure 5 $\qquad\square$

**Theorem:** 2-FWL$_L$ has stronger link discriminating power than 2-WL$_L$.

*Proof.* The proof is the same as the proof of Theorem 3.2 except that (23)-(30) works for $p, q$ and their neighbors instead of all nodes. $\qquad\square$

## C  EXTENDED DISCUSSION ON LABELING TRICK

In this section, we compare the link discriminating power between 2-WL tests and 1-WL with labeling tricks. There are two most classic labeling tricks for link prediction: the 0/1 labeling and distance-based labeling, the former labels the target nodes pair with one and other nodes with zero. A classic instance of distance-based labeling is DRNL (Double-Radius Node Labeling) in (Zhang & Chen, 2018). It constructs an injective function of distances from current node to two target nodes. Such a technique inherently makes use of the information of all paths to the target nodes within the extracted subgraph, which itself is a strong heuristic of link prediction. Note that node labeling can be directly included in label (feature) $l$.

Here we mainly discuss 0/1 labeling in the following and leave the discussion on distance-based labeling tricks and more general ones to the future work. Zhang et al. (2021) has discussed the

theoretical power of 0/1 labeling trick and showed that it enhances 1-WL's link discriminating power. We further compare the link discriminating power of 1-WL with labeling trick and 2-WL tests in the following theorem:

**Theorem C.1.** *For 0/1 labeling trick $\mathcal{L}$, 1-WL test with $\mathcal{L}$ and local 2-FWL test both do not have equal or stronger link discriminating power than the other.*

*Proof.* $(C, D)$, $(C', D')$ in Figure 5 present an example that 2-FWL$_L$ can discriminate but 1-WL with $\mathcal{L}$ cannot. On the other hand, let's consider 4-order magic square graphs. Below are two $4 \times 4$ grid graphs without node features. Each node has a number from $\{1, 2, 3, 4\}$ on it. Two nodes have edges if and only if they are 1) in the same row, or 2) in the same column, or 3) holding the same number. The colored node pair $(p, q)$, $(p', q')$ are the target links. Notice that they are both strongly regular graphs and 2-FWL cannot discriminate the two links because any node pair with edge has two common neighbors and any node pair without edge also has two common neighbors.

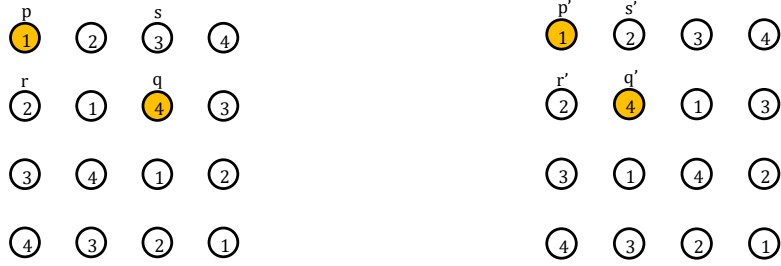

1-WL with 0/1 labeling can discriminate $(p, q)$, $(p', q')$. If not, $(r, s)$, $(r', s')$ (or $(r, s)$, $(s', r')$) will be indistinguishable from each other but distinguishable from other node pairs because they are the only nodes that have two labeled children. However they can actually be discriminated since $(r', s')$ is connected but $(r', s')$ isn't, which leads to a contradiction. $\square$

# D   MORE DETAILS ON GNN IMPLEMENTATIONS

Four proposed 2-WL variants are implemented in totally different way. First they all need to compute node embedding by 1-WL-GNN. Then after pairwisely encoding, our model mimic the 2-WL test pattern to propagate message. Our algorithms engage some tensor operation: given a 2D matrix $X$, we denote $\mathcal{A} = X^{[i,j]}$ that $X$ expands along $i^{th}, j^{th}$ dimension to 3D tensor $\mathcal{A}$. Given vector $\boldsymbol{v}, \mathcal{B} = \boldsymbol{v}^{[k]}$ denote that $\boldsymbol{v}$ expands along $k^{th}$ dimension to 3D tensor $\mathcal{B}$. Function $sum^{[i]}(X)$ means summing $X$ over its $i^{th}$ dimension, and $(\mathbf{sp})\mathbf{mm}(A, B)$ means slice-wise (sparse) matrix multiplication of $A, B$ on their first and second dimension. "|" denotes concatenation. Below we formally present our algorithms:

**Computing infrastructure.** We leverage Pytorch Geometric V2.0.2 and Pytorch V1.10.0 for model development. We train our models, measure AUROC and the inference time on an A40 GPU with 48GB memory on a Linux server.

**Baselines.** For AUROC of methods: MF, N2V, VGAE, WLNM, SEAL on non-featured datasets, we directly use the results in Zhang & Chen (2018). For AUROC of methods: VGAE, S-VGAE, TLC-GNN, SEAL, NBFNet on citation datasets, we directly use the results in Zhu et al. (2021). For performance of KGC methods: R-GCN, GraIL, INDIGO on KG datasets, we use the results in Liu et al. (2021)

**Hyperparameter tuning.** Hyperparameters are selected based on validation set performance. The best hyperparameters can be found in our code in the supplement material. Learning rate $lr$ is chosen from: $\{5e-2, 1e-2, 5e-3, 1e-3, 5e-4\}$, hidden dimension for 1-WL-GNN $h_1$: $\{32, 64, 96, 128\}$, number of hidden layers for 1-WL-GNN $l_1$: $\{1, 2, 3\}$, number of hidden layers for 2-WL-GNN $l_2$: $\{1, 2, 3\}$, hidden dimension for 2-WL-GNN $h_2$: $\{16, 24, 32, 64, 96\}$, dropout ratio for embedding layer $dp_1$, 1-WL layer $dp_2$, 2-WL layer $dp_3$: $\{0.1, 0.2, 0.3, 0.4, 0.5\}$. We use Optuna (Akiba et al., 2019) to perform random searching for hyperparameters.

---

**Algorithm 1** 2-WL Forward Network

---

**Input:** Node embedding: $\mathcal{N} \in \mathbb{R}^{n \times d}$, Adjacent matrix: $A \in \mathbb{R}^{n \times n}$, Number of layers: $L$, Target link: $(p, q)$.

1: $X^{(0)} = (\mathcal{N}^{[1,3]} \otimes \mathbf{1}_n^{[2]}) \odot (\mathcal{N}^{[2,3]} \otimes \mathbf{1}_n^{[1]})$
2: **for** $l = 0 \ to \ L - 1$ **do**
3:    $X' = mlp_{1,l}(X^{(l)} | A^{[1,2]})$
4:    $X^{(l+1)} = mlp_{2,l}\big(X^{(l)} | sum^{[1]}(X')^{[2,3]} \otimes \mathbf{1}_n^{[1]} | sum^{[2]}(X')^{[1,3]} \otimes \mathbf{1}_n^{[2]}\big)$
5: **end for**
6: $Y_{p,q} = X_{p,q,:}^{(L)} \cdot X_{q,p,:}^{(L)}$
7: **return** $Y_{p,q}$

---

**Algorithm 2** 2-WL$_L$ Forward Network

---

**Input:** Node embedding: $\mathcal{N} \in \mathbb{R}^{n \times d}$, Number of layers: $L$, Observed edge index: $\{(p_i, q_i)\}_{i=1}^m$ Target links: $\{(p_j, q_j)\}_{j=m+1}^{m+m'}$.

1: **Compute** $A^{(1)} \in [0, 1]^{n \times n}$ s.t. $A_{ij}^{(1)} = 1 \iff p_i = p_j, i \leq m$
2: **Compute** $A^{(2)} \in [0, 1]^{n \times n}$ s.t. $A_{ij}^{(2)} = 1 \iff q_i = q_j, i \leq m$
3: **for** $l = 0 \ to \ L - 1$ **do**
4:    $X^{(l+1)} = \text{GCN}_{1,l}(X^{(l)}; A^{(1)}) + \text{GCN}_{2,l}(X^{(l)}; A^{(2)})$
5: **end for**
6: $Y = mlp(X^{(L)})$
7: **return** $Y_{m:m+m'}$

---

**Algorithm 3** 2-FWL Forward Network

---

**Input:** Node embedding: $\mathcal{N} \in \mathbb{R}^{n \times d}$, Adjacent matrix: $A \in \mathbb{R}^{n \times n}$, Number of layers: $L$, Target link: $(p, q)$.

1: $X^{(0)} = (\mathcal{N}^{[1,3]} \otimes \mathbf{1}_n^{[2]}) \odot (\mathcal{N}^{[2,3]} \otimes \mathbf{1}_n^{[1]}) | A^{[1,2]}$
2: **for** $l = 0 \ to \ L - 1$ **do**
3:    $X' = mlp_{1,l}(X^{(l)})$
4:    $X'' = mlp_{2,l}(X^{(l)})$
5:    $X^{(l+1)} = mlp_{3,l}(x^{(l)} | \mathbf{mm}(X', X''))$
6: **end for**
7: $Y_{p,q} = X_{p,q,:}^{(L)} \cdot X_{q,p,:}^{(L)}$
8: **return** $Y_{p,q}$

---

**Algorithm 4** 2-FWL$_L$ Forward Network

---

**Input:** Node embedding: $\mathcal{N} \in \mathbb{R}^{n \times d}$, Adjacent matrix: $A \in \mathbb{R}^{n \times n}$, Number of layers: $L$, Observed edge index: $\{(p_i, q_i)\}_{i=1}^m$ Target link: $(p, q)$.

1: **Compute** sparse matrix: $W \in \mathbb{R}^{n \times n \times d}$ s.t. $W_{p_i, q_i} = \mathcal{N}_{p_i} \odot \mathcal{N}_{q_i}$
2: $X^{(0)} = mlp_0(W)$
3: **for** $l = 0 \ to \ L - 1$ **do**
4:    $W' = mlp_{1,l}(W)$
5:    $X' = \mathbf{spmm}(X^{(l)}, W')$
6:    $X^{(l+1)} = mlp_{2,l}(X')$
7: **end for**
8: $Y_{p,q} = mlp_3(X_{p,q,:}^{(L)} | \mathcal{N}_p \odot \mathcal{N}_q)$
9: **return** $Y_{p,q}$

---

