# OpenReview forum: "Two-Dimensional Weisfeiler-Lehman Graph Neural Networks for Link Prediction"
_ICLR.cc/2023/Conference — Submitted to ICLR 2023_

### Official Review · Reviewer_aR3Z · 2022-10-20

**Confidence:** 4
**Correctness:** 4
**Technical Novelty And Significance:** 1
**Empirical Novelty And Significance:** 2
**Recommendation:** 5

**Clarity, Quality, Novelty And Reproducibility:**

# Clarity:
- The theoretical results, examples and main arguments are clear and easy to follow.
# Quality:
- The experimental protocol and the breadth of comparison is very good.
# Novelty:
- The approach is not novel, as 2-WL-based GNNs have been proposed in the literature prior. Their use for explicit link prediction is relatively less explored, but this does not make using these models in the link prediction context a sufficiently novel modelling contribution.

# Reproducibility:
- The results in the paper appear to be largely reproducible.


**Strength And Weaknesses:**

# Strengths:
- The idea of using pair-wise node representations is intuitive and powerful, and is a natural choice for link prediction tasks.
- The empirical results and reported runtimes are convincing: The models all perform strongly, and the local variants, despite being theoretically weaker, achieve very good performance with a fraction of the running time required by labelling or sub-graph method, and this due to 2-WL-GNNs' ability to batch predictions and make them jointly.
- The paper is well-written and the preliminaries are results are well-presented. The figures are also helpful to clarify the main arguments proposed in the paper.

# Weaknesses:
- I have concerns about the novelty of the approach, as 2-WL-GNNs seem very similar to k-GNN by Morris et al. (cited in the paper). Of course, the k-GNN paper does not evaluate on link prediction tasks, but the models appear to be conceptually identical. Therefore, I do not believe that the paper makes any major novel contribution on the model development side. The main contribution instead lies in running the evaluation with multiple variants and looking at model performance from this perspective: In light of this, I would recommend rephrasing the paper's findings along these lines.
- The theoretical contribution of the paper also appears insufficient. Indeed, the results about 2-FWL versus 2-WL are relatively easy to prove, and 2-WL versus 1-WL, though interesting as it contrasts with the equivalence at the node level, is also simple. Hence, the paper overall does not provide any major novel insights on the theoretical side, either.
- Though this approach enables batched predictions, and thus offers efficiency improvements relative to sub-graph and labelling-based link prediction techniques, I believe it is misleading to present this approach as scalable in general. Indeed, local 2-WL efficiency depends explicitly on edge density and graph size, and will not be efficient on large graphs. By contrast, the sub-graph and node labelling methods impose a high overhead and cannot be batched, but can naturally scale to larger graphs, as they consider a relatively fixed-size neighborhood bounded by a fixed number of hops. Hence, the point made about efficiency is only really true for smaller graphs. The true situation appears to be a trade-off: On smaller graphs, sub-graph methods are heavily redundant and the lack of batching is very sub-optimal, making local 2WL a clearly more efficient choice. However, on larger graphs, the redundancies of sub-graph methods are relatively reduced and batching is no longer practically viable. In turn, these methods become more promising from a computational complexity perspective. I thus strongly suggest that these findings be included in the paper and discussed holistically.

**Summary Of The Paper:**

The paper proposes a 2-dimensional WL GNN to be used for link prediction, based on global and local variants of the 2-WL graph isomorphism heuristic. In particular, the paper motivates the use of link-level representations, as opposed to node-level representations in standard 1-WL-based GNNs, so as to directly make inferences for link prediction. To this end, the paper introduces the standard 2-WL algorithm and its more powerful folklore counterpart (2-FWL), as well as their local relaxations 2-WL_L and 2-FWL_L respectively, which only consider existing edges in the graph. The paper then establishes expressive power results among all four variants and standard 1-WL, and shows that, for link discrimination, 1) 2-FWL is stronger than 2-WL, 2) 2-WL is stronger than 1-WL, 3) 2-WL_L is equivalent to 1-WL, and 4) 2-FWL_L is stronger than 1-WL but weaker than 2-FWL. Building on these results, the paper takes inspiration from existing literature, namely k-GNNs and PPGNs, to propose an 2-WL-GNN architecture. More specifically, a standard 1-WL GNN is first run, with its representations then pooled to obtain link-level representations that are subsequently refined by a 2-WL layer. In turn, GNN layers corresponding to each of 2-FWL, 2-WL, and 2-FWL_L are provided. Finally, the paper runs an extensive empirical analysis of the different proposed variants and compared against several state-of-the-art GNNs for link prediction, and shows that 1) 2-WL-GNN models perform strongly, mostly surpassing their competitor models, and 2) achieve this performance with a much smaller computational overhead.

**Summary Of The Review:**

All in all, I think that the paper's use of 2-WL-GNNs for link prediction is intuitive and interesting. However, the ideas brought forward in this paper are not novel, but are rather better seen as an evaluation of 2-WL-GNNs from a link prediction perspective. The empirical findings are comprehensive and convincing, but also do not produce any novel insights or analyses that shed new light on our understanding of GNNs for link prediction. Hence, on the balance, I lean towards a weak reject for the paper, as it makes insufficient novel contributions towards producing new techniques for link prediction with GNNs. However, I am happy to change my verdict should the authors address my above concerns.

---

> ### Author Response · Authors · 2022-11-09
> **Author Response**
>
> We thank the review for reading our paper and giving honest and valuable feedback. We respond them as follows.
>
> - “Weakness 1”
>
> Our works has notable difference from [1]. Apart from the link prediction tasks, our three models local 2-WL, 2-FWL, local 2-FWL are totally different from models in [1]. Our definition of local version is different from [1] as well. We delicately chose our own local version to meet both low complexty and strong expressive power in link prediction context. Our proposal and implementation of local 2-FWL is a complete novelty with no similar models in any related field. It must fullfill strong theoretic power, low complexity, and feasibility to realise it. The formal expression of local 2-FWL greatly reduce the space complexity while remain most of 2-FWL’s distinguish power. The implementation uses complex sparse matrix operation to maximally fullfill such expressive power. All of these are the distinction of our work compared to others.
>
> - “Weakness 2”
>
> The theoretic results of link-level seems to be the same as those of node-level, but this still need rigorous proof. Additionally, our link-level WL test is different from original one in another way that ours doesn’t contain an aggregation of all units in the last step, due to the nature of locality in link prediction context. These differences make us unable to directly use existed theoretic results, nor to use classic proving techniques like pebbling games. Furthermore, our focus is not on the theoretic property of link-level 2-WL which only occupies one introducting passage. We also did extensive experiment on different tasks with time complexity analysis, and many design and compromise are contained in model inplementation. It is our major concern that how 2-WL based models performs on link prediction problem in practice and how to find a balance between time/space complexity and expressive power.
>
> - “Weakness 3”
>
> It is true that 2-WL based methods have scalability problem even if its batch prediction performs well on graphs of relatively small size. We clarified this weakness in our Limitation subsection. Actually our goal is to neutrally discuss the performance of 2-WL based method on link prediction problems. It is unavoidable that 2-WL model must have scalability issue compared to those subgraph extracting method. But we also aim to find whether the 2-WL variants has some specific advantages over those labeling method, and how these points could illuminate thoughts and insights. For example, as for the batch prediction of our model, a natural question is, what fundamental property brings about such merit? It is a kind of scheme or symmetry in the model that all of the links, whether questioned or not, are treated the same. Once a model possesses such feature it could achieve high efficiency in training/inference process. Can we find more such models for link prediction? This is also an interesting question to be transferred to other fields like knowledge graph completion, matrix completion, etc. Above all, there’s need to further highlight the limitation of this method and our motivation of exploration and to downplay the achievement on small dataset. We will address this in our revision.
>
> [1] Weisfeiler and Leman go sparse: Towards scalable higher-order graph embeddings (Preprint arXiv:1904:01543) Christopher Morris, Gaurav Rattan, Petra Mutzel, Neural Information Processing Systems (NeurIPS) 2020.

---

> > ### Comment · Reviewer_aR3Z · 2022-11-19
> > **Reviewer Response**
> >
> > Thank you for your rebuttal. I address your comments below:
> >
> > - Weakness 1: I am unfortunately not convinced. You may indeed use different variants of 2-WL with efficiency in mind, but this alone is not sufficiently novel. The idea of local 2-WL has also been explored previously. I suggest you better quantify this claim.
> >
> > - Weakness 2: I understand that the theory is not a primary concern. However, this is a major unclear aspect of the work, which I feel must be better addressed to improve the contribution of the paper. I accept the authors' view that the prioritization of efficiency is their motivation. However,, the fact that their approach is not sufficiently distinguished from other 2-WL variants save for this efficiency focus, corroborated by a lack of theoretical underpinning for this approach, make it hard to support the paper at the moment.
> >
> > - Weakness 3: Thank you for your constructive discussion. I look forward to seeing a more balanced presentation of this aspect in the paper.

---

### Official Review · Reviewer_kWUU · 2022-10-21

**Confidence:** 3
**Correctness:** 4
**Technical Novelty And Significance:** 2
**Empirical Novelty And Significance:** 2
**Recommendation:** 5

**Clarity, Quality, Novelty And Reproducibility:**

The paper is easy to read, although it suffers from quite a few typos. The ideas presented in the paper are quite incremental and not very novel. That is, they more or less straightforwardly apply the 2-WL for the problem of link prediction, and not, e.g., graph classification as in previous works. The authors provided the source code and the experimental details are sufficiently described.

**Strength And Weaknesses:**

**Strengths**
- Simple approach that sometimes works better than baseline approaches

**Weaknesses**
- Presentation needs to be improved
- Incremental paper
- Theoretical results are rather obvious


**Remarks/Suggestions**
- The definition in Eq. 2 is slightly confusing as you use j for indices as well as vertices.
- In section 2.1, you only seem to define the neighborhoods of k-tuples but not the actual algorithm. Hence, readers not familiar with k-WL might have a hard time.
- In section 2.2, you are not defining the initial coloring of the tuples. That is, the original k-WL uses the atomic type which is crucial for its expressivity.
- The definition of the local variations is quite close to [1]. That is, [1] requires that the exchanged nodes are adjacent while you require that the non-replaced node and the replacing node are adjacent. Further, note that all your neighboring nodes are edges, non-edges are not considered.
- Figure 3 hinges on the fact the number of nodes of the graphs is different. You might want to use a pair with the same number of nodes.
- In section 3.2, a lower bound on the time and memory complexity of the local version is still n^2. You might want to make this more clear in the paper.
- In section 4.1, it is not clear if the proposed neural architectures indeed possess the same expressive power as the 2-WL. Only a heuristic argument is given.
- In table 5, for the running time comparison, you only use small-scale datasets, e.g., CORA. It would be interesting to use larger datasets.
- Theorem 3.1, 3.2, and 3.4 can be proved via simple counterexamples. The pursued proof strategy via unravelings/unrollings seems overly complicated.

**Questions**
- Can you quantify if your local k-WL is more expressive than the local 2-WL defined in [1]?
- What are the benefits of using the local 2-WL if it has the same expressive power as the 1-WL?
- Why don't you prove Theorem 3.1, 3.2, 3.4 via simple counter examples? It seems very plausible that it follows directly from a CFI-like construction for k=2.

**Minor points/Typos**
- In section 2.1, first paragraph "k-dimensional WL test" -> "The k-dimensional WL test", also second paragraph
- In section 2.2, there is a broken reference
- In section 3.1, "the Folklore" -> "the folklore"
- Section 6.2, "we" -> "We"

[1] Weisfeiler and Leman go sparse: Towards scalable higher-order graph embeddings (Preprint arXiv:1904:01543)
Christopher Morris, Gaurav Rattan, Petra Mutzel,
Neural Information Processing Systems (NeurIPS) 2020.


**Summary Of The Paper:**

The paper deals with the problem of link prediction in graphs. Specifically, it proposes to leverage variations of the 2-dimensional Weisfeiler-Leman algorithm (2-WL) for this problem.  The authors show how the 2-WL and folklore 2-WL (2-FWL) can be leveraged straightforwardly to design GNN-like neural architectures that learn representations for edges and non-edges. Moreover, similarly to [1], they define a local version that only considers a subset of all k-tuples during aggregation.

Theoretically, they compare the above variants in terms of expressive power. That is, they show that the 2-FWL is strictly strong than the 2-WL in distinguishing links. Further, they show that the local variant has the same expressive power as the 1-WL.

Empirically, the proposed architectures are evaluated on standard link prediction and knowledge completion datasets, showing somewhat promising performance.

**Summary Of The Review:**

This is a solid, incremental paper. The theoretical results are not very surprising and the experimental results are mixed, not clearly indicating why 2-WL-based methods are superior for link prediction problems.

---

> ### Author Response · Authors · 2022-11-09
> **Author Response**
>
> We appreciate that the reviewer reading our paper and give honest feedback. We answer the mentioned questioned in the below.
>
> - Re “Q1: Can you quantify if your local $k$-WL is more expressive than the local 2-WL defined in [1]?”
>
> First of all, we only defined our local 2-WL and 2-FWL, and such definition cannot be extended to $k\geqslant 2$. The local $k$-WL defined in [1] can be applied to arbitrary $k$.
>
> Our local 2-WL is strictly weaker than that in [1]. All of information in ours is covered by local 2-WL in [1], and ours cannot capture three node structure like common neighbors while [1] can. However such distinguishing power comes at a price that the space complexity of the second model on connected graph is $O(n^2)$, compared to $O(m)$ of our local 2-WL.
>
> - Re “Q2: What are the benefits of using the local 2-WL if it has the same expressive power as the 1-WL?”
>
> It doesn’t present benefit from theoretic view. It is more of empirical value that we discover whether such pairwise encoding model works for link prediction. Our main purpose is exactly to discuss those 2-WL based method performance on link, and how to deal with the trade off between time/space complexity and expressive power. Local 2-WL also serves as a comparison method with other 2-WL variants to demonstrate if the experiement reflects theoretic results. A similar model is studied in recent work on knowledge graph compeletion [2].
>
> - Re “Q3: • Why don't you prove Theorem 3.1, 3.2, 3.4 via simple counter examples? It seems very plausible that it follows directly from a CFI-like construction for $k$=2.”
>
> The CFI graph is true counterexample for $k$=2, but the other side still needs to be proved. But rigorously speaking, CFI graph is example for graph-level test, which still need to be combined with certain link. Therefore we directly display concrete examples that are also simple enough. Additionally, our link-level WL test is different from original one in that ours doesn’t contain an aggregation of all units in the last step, due to the nature of locality in link prediction context. These differences make us unable to directly use early theoretic results, nor to use classic proving techniques like pebbling games.
>
> [1] Weisfeiler and Leman go sparse: Towards scalable higher-order graph embeddings (Preprint arXiv:1904:01543) Christopher Morris, Gaurav Rattan, Petra Mutzel, Neural Information Processing Systems (NeurIPS) 2020.  [2] Shuwen Liu, Bernardo Grau, Ian Horrocks, and Egor Kostylev. Indigo: Gnn-based inductive knowledge graph completion using pair-wise encoding. NIPS, 2021.

---

### Official Review · Reviewer_5pff · 2022-10-25

**Confidence:** 4
**Correctness:** 3
**Technical Novelty And Significance:** 1
**Empirical Novelty And Significance:** 2
**Recommendation:** 3

**Clarity, Quality, Novelty And Reproducibility:**

The paper has many typos and ill-formed sentences, and in some cases, it is really hard to parse. It would benefit from a thorough pass. In terms of novelty, I find the work rather incremental. The experimental results largely align with intuition and it is good to see that these algorithms lead to improvements, but they are subject to similar (and maybe worse) scalability issues than, e.g., other approaches (NBFNets are somewhat scalable on KGs, though not really on heterogenous graphs).

**Strength And Weaknesses:**

Strengths:

- GNNs are still relatively weak on link-level tasks (esp., on knowledge graphs) compared to node-level tasks and there is a strong need for further research in this context.
- The expressiveness limitations of GNNs are well-understood, but such limitations are more severe on link-level tasks, since the class of pair-wise representations that cannot be distinguished by 1-WL learned features is quite large.

Weaknesses:

- Many of the important points raised in this paper are taken from existing literature, and the authors should do a better job at locating this work in reference to existing works they cite, including [1-4]. The goal of this work is different, but the overlap between these papers  is surprisingly large, which makes it really hard to delineate the exact contribution of this work.

- The algorithms proposed are very close to the algorithms from $k$-GNNs (Morris et al, 2019) are their variations proposed in the literature. The authors also use 1-WL algorithms to learn initial features, and then run the 2-WL variants, which is basically a special case of the idea of using 1-2-3-GNNs from (Morris et al, 2019).

-  Most results are straightforward: $k$-FWL ($=k+1$-WL ) is more expressive than $k$-WL already on node-wise representations, and so this clearly translates to pair-wise representations, making Theorem 3.2 straightforward (and one can actually use this correspondence to show the result). Theorem 3.4  follows the same ideas as Theorem 3.2. Theorem 3.1 is more interesting, which pinpoints the limitations of 1-WL on links, and shows an example which separates 1WL from 2WL on link-level tasks, but then, similar examples are already given in earlier works, so this seem appears also very incremental.

- There are also some technical concerns regarding how WL and FWL algorithms are defined: To my understanding, WL algorithms also aggregate over non-neighbours (even 1-WL); see, e.g., "Martin Grohe, The logic of graph neural networks, LICS 2021". While this doesn't seem to make a difference on single graphs, it does make a difference when we consider sets of graphs, as noted in "Barcelo et al., The logical expressiveness of graph neural networks, ICLR 2020".


[1] Muhan Zhang and Yixin Chen. Weisfeiler-lehman neural machine for link prediction. KDD 2017. (this is cited twice with different structure)
[2] Muhan Zhang and Yixin Chen. Link prediction based on graph neural networks. NeurIPS, 2018.
[3] Muhan Zhang and Yixin Chen. Inductive matrix completion based on graph neural networks. ICLR, 2020.
[4] Muhan Zhang, Pan Li, Yinglong Xia, Kai Wang, and Long Jin. Labeling trick: A theory of using graph neural networks for multi-node representation learning. NeurIPS, 2021.


**Summary Of The Paper:**

The main idea of this paper is to design dedicated models for link & relation prediction: Standard graph neural networks which learn node-wise representations are also used for link-level tasks, but their lack of expressive power is more severe on link-level tasks. One way of alleviating this limitation is to apply a so-called labelling trick, which usually incurs a large computational overhead. This paper proposes directly learning pair-wise representations, grounded in 2-WL and its local variants. Experiments are conducted both on link prediction and on knowledge graph completion benchmarks.

**Summary Of The Review:**

In its current state, the paper looks incremental on existing work, and the presentation does not help much in identifying the precise contributions. There are also some concerns outlined in the review, which leads me to suggest a weak reject.

---

> ### Author Response · Authors · 2022-11-09
> **Author Response**
>
> We thank the reviewer for reading our paper and pointing out some concerns. We answer tham as follows.
>
> - Re “Weakness 1: Many of the important points raised in this paper are taken from existing literature, and the authors should do a better job at locating this work in reference to existing works they cite, including [1-4]. The goal of this work is different, but the overlap between these papers is surprisingly large, which makes it really hard to delineate the exact contribution of this work.”
>
> [1]-[4] are all link prediction methods, but they have notable distinction from our model. By using 1-WL coloring, [1] order \textbf{nodes} to construct the unique adjacency matrix as graph encoding.  [2], [4] applied GNN on original graph with \textbf{node} labeling to highlight the target link and its surrounding structure. [3] transplanted GNN with node labeling on matrix completion problems. Our method doesn’t contain any node ordering or labeling technique, and also apply GNN on the two-order graph instead of original one. Thereby our method have high speed since extracting subgraph is needless and these are significant differences from works of referrence. [1] is mistakenly referred twice which we will correct in the revision.
>
> - Re “Weakness 2: The algorithms proposed are very close to the algorithms from $k$-GNNs (Morris et al, 2019) are their variations proposed in the literature. The authors also use 1-WL algorithms to learn initial features, and then run the 2-WL variants, which is basically a special case of the idea of using 1-2-3-GNNs from (Morris et al, 2019).”
>
> Our vanilla 2-WL model is actually similar to (Morris et al, 2019), but the other three models local 2-WL, 2-FWL, local 2-FWL are totally different, which are also prioritized models in this work. Our definition of local version is different from (Morris et al, 2019) as well. Our proposal and implementation of local 2-FWL is a complete novelty with no similar models in any related field. The  formal expression of local 2-FWL greatly reduce the space complexity while remain most of 2-FWL’s distinguish power. The implementation uses complex sparse matrix operation to maximally fullfill such expressive power. All of these are among the distinction from our work to Morris et al.
>
> - Re “Weakness 3: Most results are straightforward: $k$-FWL (=$k$+1-WL ) is more expressive than $k$-WL already on node-wise representations, and so this clearly translates to pair-wise representations, making Theorem 3.2 straightforward (and one can actually use this correspondence to show the result). Theorem 3.4 follows the same ideas as Theorem 3.2. Theorem 3.1 is more interesting, which pinpoints the limitations of 1-WL on links, and shows an example which separates 1WL from 2WL on link-level tasks, but then, similar examples are already given in earlier works, so this seem appears also very incremental.”
>
> The theoretic results of link-level seems to be the same as those of node-level, but this still need rigorous proof. Additionally, our link-level WL test is different from original one in another way that ours doesn’t contain an aggregation of all units in the last step, due to the nature of locality in link prediction context. These differences make us unable to directly use early theoretic results, nor to use classic proving techniques like pebbling games. Furthermore, our focus is not on the theoretic property of link-level $2$-WL. Instead it serves to our major concern that how 2-WL based models performs on link prediction problem in practice and how to find a balance between time/space complexity and expressive power.
>
> - Re “Weakness 4: There are also some technical concerns regarding how WL and FWL algorithms are defined: To my understanding, WL algorithms also aggregate over non-neighbours (even 1-WL); see, e.g., "Martin Grohe, The logic of graph neural networks, LICS 2021". While this doesn't seem to make a difference on single graphs, it does make a difference when we consider sets of graphs, as noted in "Barcelo et al., The logical expressiveness of graph neural networks, ICLR 2020””
>
> We have noticed this point as well. There’s actually some literature that define 1-WL in the way that non-neighbors are included in each coloring refinement. They are of the same power as the “local” one because the latter aggregate all node color to represent the whole graph in the last step. It can be proved that there’s no need to consider non-neighbors in previous steps. This 1-WL definition can be found in [5] on which we depend. Of course, such difference only appears in $k$=1 circumstances.

---

> > ### Author Response · Authors · 2022-11-09
> > **Author Response**
> >
> > [1] Muhan Zhang and Yixin Chen. Weisfeiler-lehman neural machine for link prediction. KDD 2017. [2] Muhan Zhang and Yixin Chen. Link prediction based on graph neural networks. NeurIPS, 2018. [3] Muhan Zhang and Yixin Chen. Inductive matrix completion based on graph neural networks. ICLR, 2020. [4] Muhan Zhang, Pan Li, Yinglong Xia, Kai Wang, and Long Jin. Labeling trick: A theory of using graph neural networks for multi-node representation learning. NeurIPS, 2021.[5] Keyulu Xu, Weihua Hu, Jure Leskovec, and Stefanie Jegelka. How Powerful are graph neural networks? ICLR, 2019.

---

### Decision · Program_Chairs · 2023-01-20

**Decision:**

Reject

**Justification For Why Not Higher Score:**

Insufficient novelty.

**Justification For Why Not Lower Score:**

N/A

**Metareview: Summary, Strengths And Weaknesses:**

The paper addresses the problem of link prediction using a GNN architecture based on a more expressive 2-dimensional Weisfeiler-Lehman graph isomorphism test (2-WL). 2WL-equivalent GNN architectures have been considered before (notably by Morris and Maron), so the novelty of the paper is limited. This view is shared after the reviewers, also after the rebuttal. While the paper is interesting, we believe novelty or experimental results are below the bar. We recommend rejection.